# Characteristics of the Skin Microbiome in Selected Dermatological Conditions: A Narrative Review

**DOI:** 10.3390/life12091420

**Published:** 2022-09-12

**Authors:** Esther Olunoiki, Jacqueline Rehner, Markus Bischoff, Elena Koshel, Thomas Vogt, Jörg Reichrath, Sören L. Becker

**Affiliations:** 1Institute of Medical Microbiology and Hygiene, Saarland University, 66421 Homburg, Germany; 2“Solution Chemistry of Advanced Materials and Technologies” (SCAMT) Institute, ITMO University, 191002 St. Petersburg, Russia; 3Department of Dermatology, Venereology, Allergology, Saarland University Medical Center, 66421 Homburg, Germany

**Keywords:** dermatology, human microbiome, metagenomics, microbial diversity, psoriasis, candidiasis, hidradenitis suppurativa

## Abstract

The skin is the largest and outermost organ of the human body. The microbial diversity of the skin can be influenced by several variable factors such as physiological state, lifestyle, and geographical locations. Recent years have seen increased interest in research aiming at an improved understanding of the relationship between the human microbiota and several diseases. Albeit understudied, interesting correlations between the skin microbiota and several dermatological conditions have been observed. Studies have shown that a decrease or increase in the abundance of certain microbial communities can be implicated in several dermatological pathologies. This narrative review (i) examines the role of the skin microbiota in the maintenance of skin homeostasis and health, (ii) provides examples on how some common skin diseases (acne inversa, candidiasis, psoriasis) are associated with the dysbiosis of microbial communities, and (iii) describes how recent research approaches used in skin microbiome studies may lead to improved, more sensitive diagnostics and individual therapeutics in the foreseeable future.

## 1. Introduction

The skin is the external and the largest organ of the human body that functions as an important physical barrier (e.g., to protect against harmful external conditions and invasion of pathogens) and communication tool (e.g., to respond to external stimuli) [1,2,3]. The human skin is home to millions of microorganisms, which are referred to as skin microbiota [1,2,3]. When compared to the more extensively studied gut microbiota, the skin microbiota seem to have similar characteristics, e.g., they also confer protection against pathogens, maintain the skin homeostasis, stimulate, and activate various immune responses, metabolize natural products, and produce antimicrobial peptides [3,4,5,6]. The physiological composition of the skin can be divided into four major micro-environments, i.e., (1) sebaceous (face, chest, and back), (2) moist (elbow, knees, genitalia), (3) dry (palms), and (4) foot-specific [3,5,7]. These micro-environments are characterized by specific microbiota that may have a considerable impact in maintaining a ‘healthy’ skin, whereas microbiota ‘shifts’ can facilitate the development and progression of diseases [3,5,8,9]. Members of the genus *Cutibacterium* (previously known as *Propionibacterium*), for example, dominate the sebaceous (oily) sites of the human skin, while the moist sites are primarily colonized by members of the genera *Staphylococcus* and *Corynebacterium*, and the female genital tract is characterized by a predominance of *Lactobacillus* [3,8,10,11]. Although not as diverse as the bacterial community, viral and fungal commensals are also present on our skin; for example, fungi of the genus *Malassezia* are widely distributed across the sebaceous sites of the skin [3,8,12,13].

The composition of the skin microbiota is assumed to be relatively stable. However, an imbalance in the proportion of ‘normal’ skin flora to pathogenic microorganisms, a phenomenon known as dysbiosis, results in the initiation and progression of several dermatological diseases [1,3,8]. Until recently, microbiological culture-based methods were primarily used for the isolation and characterization of skin microbial communities [1,3,8,14]. However, these methods underestimate the ‘true’ diversity of organ-specific microbiota because many bacterial species differ considerably with regard to nutritional requirements and growth rates on agar plates, and a number of species simply cannot be grown in culture [3,8,14,15,16,17,18]. *Staphylococcus* spp., for example, grows faster than *Cutibacterium* spp. and *Corynebacterium* spp. on agar plates inoculated by skin swabs, and the latter may thus be overlooked in culture-based studies. In an attempt to circumvent these biases and capture the total diversity of the skin microbiome, sequencing approaches are nowadays used to identify members of the skin microbial communities [1,3,15,18]. A commonly employed approach makes use of the targeted amplification of ‘molecular fingerprints’ of specific pathogen classes, e.g., the housekeeping 16S ribosomal RNA gene for bacterial communities [3,8,17] and the internal transcribed spacer 1 (ITS1) region of the eukaryotic ribosomal gene for fungi [3,8,15,16], which is then followed by nucleic acid sequencing of the amplified product for species identification. However, the constant evolution of sequencing technologies from conventional Sanger sequencing to more high-throughput techniques such as pyrosequencing and Illumina sequencing allows for an improved, more precise microbiota characterization, which is facilitated by increased read depths and shorter read lengths [3,8,19,20]. Amplicon sequencing, i.e., the sequencing of previously amplified nucleic acids from specific microbiota, can capture genetic variability in microbial communities efficiently [1,3,21,22,23]. Such amplicons can then be analyzed using different software tools such as mothur [3,24] and ‘Quantitative Insights Into Microbial Ecology’ (Qiime) [25]. However, amplicon sequencing is prone to different shortcomings and may cause an analytical bias, which might be overcome by the application of the more recent shotgun metagenomic sequencing [1,2,3,8]. Through shotgun metagenomics sequencing, all the genetic materials present in a given sample are analyzed simultaneously by untargeted (and unbiased) sequencing of nucleic acids without prior polymerase chain reaction (PCR) assays to amplify specific sequences. Many short sequences are being generated, which are then reconstructed by computational biology into consensus sequences [8,26]. Shotgun metagenomics provides sufficient information to differentiate microbial genetic materials into species and further into strains [8,21,26]. This ability for strain differentiation might be important to elucidate the functional differences within a species [8,21,26,27,28].

The aforementioned sequencing approaches have enabled researchers to depict a more accurate picture of microbiota at different body sites. Here, we review recent evidence pertaining to sequencing-based studies of skin microbial communities, including their role in health, disease initiation, and progression.

## 2. Materials and Methods

A narrative literature review on the skin microbiome in health and several selected diseases was conducted using published articles available on the PubMed/MEDLINE database. Research articles published until December 2021 were selected on the basis of the methods used and the relevance of the results. No language restriction was applied during the search.

## 3. The Role of Microbiota in Healthy Skin

Structurally, the skin is composed of three distinct layers: the epidermis, the dermis, and the subcutaneous fat tissue [29,30]. The epidermis, which is the outer layer of the skin, is differentiated into the stratum corneum, stratum lucidum, stratum granulosum, and stratum basale [3,30]. The stratum corneum, which is the outermost layer, is composed of terminally differentiated and enucleated keratinocytes that are chemically cross-linked and act as a barrier [28,29,30]. The keratinocytes present in the stratum granulosum are cysteine-histidine rich and bind keratin [28,31]. Basal keratinocytes, immune cells (Langerhans cells, T cells, and melanocytes) can be found in the stratum basale [28,31]. The dermis is located beneath the epidermis. It is differentiated into the papillary sub-layer that facilitates the transport of nutrients [3,28,32] and the reticular sub-layer in which the hair follicles, sebaceous glands, and sweat glands are present. The dermis is also home to fibroblasts, myofibroblasts, and immune cells (macrophages, lymphocytes, and mast cells) [28,31,32,33]. The subcutaneous fat layer is composed of fibrocytes and adipocytes. This layer produces a wide variety of growth factors, adipokines, cytokines, and immune cells. The adipose tissue also stores energy and functions as an endocrine gland, which is crucial for glucose homeostasis and lipid metabolism [34,35,36,37]. The skin plays a vital role in stimulating and training the human immune system [38,39,40]. Studies have shown that the microbiome is influenced by many factors, such as environment, lifestyle, diet, and medication. Changes within these factors could lead to a decrease in the diversity of the human microbiota, thereby negatively impacting the host immune system and microbe communication, which may lead to an altered immunological tolerance and miseducation of the immune system. The sebum produced by the sebaceous gland lubricates the hair and skin. Sapienic acid produced by the hydrolysis of sebum by commensal microbes acts alongside other antimicrobial peptides such as cathelicidin, beta-defensins, and antimicrobial histones to control microbial colonization [1,38,39,40]. Eccrine sweat secreted directly onto the skin surface creates unfavorable conditions for the survival and proliferation of microbes [3,38,41]. The highly colonized dermal appendage is home to a variety of microbes. This environment allows for ample interaction between microorganisms and host cells [38]. Dendritic cells are found in the epidermis and act as antigen-presenting cells to T lymphocytes in the nearby lymphoid organs. T cells become activated in the presence of invading pathogens, and trigger an inflammatory response. However, a dysregulation in this mechanism can occur, which may result in the emergence of inflammatory skin diseases [38].

Extensive microbe-microbe interaction is expected between organisms that share a similar niche as they compete for nutrients and other growth factors [38]. For example, *Staphylococcus aureus,* which is a commensal skin colonizer in roughly 30 percent of the population, has been studied extensively with regard to its interaction with other microbes. *S. aureus* is an opportunistic pathogen that can cause severe infection, particularly in immunocompromised hosts. Other *Staphylococcus* spp. that inhabit the skin of humans, for example *Staphylococcus hominis* and *Staphylococcus lugdunensis*, may produce antimicrobial peptides that specifically inhibit colonization by *S. aureus*. Other species are also known for their potentially beneficial effect on the human skin, such as *Lactobacillus* spp., which prevent pathogen colonization and produce anti-inflammatory secondary metabolites [39,40,41]. The microbial diversity of the skin microbiota is influenced by several factors such as lifestyle, age, medications, etc. [3,8,38,42,43]. The immune system often recognizes the host’s commensal microbiota and establishes a mutualistic interaction. Some bacteria, such as *Staphylococcus epidermidis*, can also interact with the host’s keratinocytes directly, thereby inducing the production of antimicrobial peptides via immune cell signaling [40]. However, dysbiosis may occur and can trigger the initiation of chronic inflammatory skin disorders [38,42,43,44]. The incidence of specific pathologies such as atopic dermatitis and psoriasis have risen considerably in high-income countries, and it is speculated that dietary changes, lifestyle modifications, hygiene, and environmental factors may have contributed to this, as they may adversely affect the microbial diversity and metabolism of the skin, which might in turn lower the organ’s immune tolerance [42,43]. Disease-specific examples will be provided in the following sections for three selected conditions, i.e., (1) acne inversa, (2) (muco-)cutaneous candidiasis, and (3) psoriasis. Psoriasis and acne inversa are not directly associated with skin infections by pathogenic microorganisms. However, psoriasis as one of the most prevalent skin conditions is being studied intensively in regard to microbiome analysis. Researchers aim to find microbial species which affect this disease, both positively, and negatively. The role between the skin microbiome and acne inversa, on the other hand, has not been established yet. In contrast to psoriasis, research on acne inversa is still in the early stages. In this review, we would like to point out the current state of research on these two diseases. In contrast to non-infectious skin diseases, we also aim to elucidate the impact of the skin microbiome during the infectious disease candidiasis.

## 4. The Role of Microbiota in Selected Dermatological Diseases

### 4.1. Hidradenitis Suppurativa (Acne Inversa)

Hidradenitis suppurativa (HS), also known as acne inversa, is a chronic inflammatory skin disorder [45,46,47]. It is clinically characterized by deep-seated fistulae, nodules, abscesses, sinus tracts, and scars in the axillae, inguinal folds, perianal and perineal regions, buttocks, and infra/inter-mammary folds. HS is a painful and discomforting skin condition, which has an adverse impact on the psycho-social health of affected patients. Indeed, research has elucidated that people suffering from HS develop depression more frequently and are twice more likely to be unemployed [48,49]. The morbidity associated with HS is significantly higher than observed in other inflammatory skin diseases [49,50]. HS severity can be measured by different staging systems, e.g., the Hurley classification, Sartorius scoring, and HS severity index [45]. With regard to epidemiology, HS has a relatively high prevalence in young females of African descent [45,51]. The etiology of HS has been linked to several contributing factors such as (i) cutaneous microbiome dysbiosis, (ii) genetics, (iii) lifestyle specificities and obesity, (iv) hormonal dysbalance, and (v) immune system modifications. Here, we focus mainly on the influence of the skin microbiota on disease severity.

A variety of studies demonstrated that skin microbiota play a key role in the etiopathogenesis of HS [52,53,54,55,56,57,58,59,60,61,62,63,64,65,66,67]. Using a bacteriological culture-based approach, Brook et al. described a polymicrobial nature of HS lesions. *S. aureus*, *Streptococcus pyogenes*, and *Pseudomonas aeruginosa* were the most prevalent aerobic bacteria, and *Peptostreptococcus* spp., *Prevotella* spp., microaerophilic streptococci, *Fusobacterium* spp., and *Bacteroides* spp. were the most common anaerobes [45,46,47,48,49,50,51,52,53,54,55,56,57]. Lapin et al. confirmed such a bacterial diversity in HS lesions and described *S. aureus* and coagulase-negative staphylococci as the most abundant aerobic bacteria species, while *Peptostreptococcus* spp. and *Cutibacterium acnes* were the most common anaerobes [68]. However, *S. aureus* was not frequently isolated from acute lesions, which raises the question of how the microbiota composition might shift during acute flares of HS and in different disease activity stages. A study carried out by Guet-Revillet et al. employed a combination of microbiological culture and metagenomic techniques to address this issue. According to the bacterial culture method, two microbiological profiles were apparent (A and B). In profile A (Hurley stage 1), *S. lugdunensis* was predominant, while in profile B (Hurley stages 2 and 3), a variety of anaerobic bacteria species were isolated. Metagenomic sequencing after PCR amplification of 16S rRNA yielded similar results, with staphylococci predominating in profile A and different anaerobic species (mainly *Prevotella* spp., *Porphyromonas* spp., *Anaerococcus* spp., and *Mobiluncus* spp.) being more characteristic of more advanced disease stages of the disease [53]. More recent studies carried out by Naik et al. showed a decrease in the abundance of *Cutibacterium* spp. in a study cohort which suffered from HS and was instructed to avoid any type of detergents, cosmetics, oral and topical antibiotics, and bathing before skin swabs were taken [68]. This result was consistent with other studies, which reported a decrease of *Cutibacterium* spp. and an increase of *Prevotella* spp. and *Porphyromonas* spp. as being potentially linked to disease progression and severity [69]. Key findings are summarized in Table 1. However, the aforementioned studies are limited by their small sample sizes, the use of PCR-based amplification before sequencing, and potential cross-contamination [68,69]. Hence, further research should address HS patients with varying disease activities by examination of skin swabs employing unbiased shotgun metagenomic sequencing to overcome the potential shortcomings of culture and amplicon sequencing.

### 4.2. (Muco-)Cutaneous Candidiasis

Candidiasis or thrush is a fungal infection caused by members of the genus *Candida* [70]. Only a few members of the genus are pathogenic to humans. Out of the over 150 species of *Candida*, *Candida albicans* is considered the most pathogenic. Yet, other species such as *C. glabrata*, *C. tropicalis*, *C. krusei*, *C. kefyr*, *C. guilliermondii*, *C. dubliniensis*, and, more recently, *C. auris*, may also cause human disease, and some of these pathogens are considered as potential public health concerns because of their relatively high propensity to cause outbreaks and to become resistant to commonly used anti-fungal agents [71,72,73,74,75,76,77,78,79,80,81,82,83,84,85,86]. *C. albicans* is most commonly found in the vagina of about 30 percent of women [87]. Other non-*C. albicans* species are also found in the reproductive tract of about 10–30 percent of women [87,88,89,90,91,92,93]. *Candida* species are opportunistic pathogens that cause infections ranging from superficial oral thrush and vaginitis to systemic, potentially fatal infections (candidemia) [70,71,94]. Mucocutaneous candidiasis can be classified into non-genital (oropharyngeal and invasive infections) and genitourinary diseases [76]. Here, we will focus on the common genitourinary diseases caused by *Candida* species, which share many characteristics with cutaneous candidiasis. Candidiasis occurs in men, women, and children alike. The most prevalent genitourinary diseases are vulvovaginal candidiasis in women, balanitis, balanoposthitis in men, and candiduria in both sexes and children [76]. Vulvovaginal candidiasis (VVC) is a common infection of the lower female reproductive tract [76,77]. VVC is characterized by itching, burning and painful urination, redness, curd-like and foul-smelling vaginal discharge, and painful sexual intercourse [76,77]. About 75 percent of women develop VVC at least once during their lifetime [76,77,78]. VVC can be recurrent and it is medically termed recurrent vulvovaginal candidiasis (RVVC), if it occurs more than four times per year [76,77]. Approximately 8 percent of women globally suffer from RVVC [76,77]. Several factors influence the occurrence of VVC and RVVC such as the use of antibiotics, sexual activity and HIV infection, high estrogen-containing oral contraceptives, pregnancy, the use of sodium cotransporter 2 (SGLT2) inhibitors, and uncontrolled diabetes mellitus [76,80]. Compared to oropharyngeal candidiasis, VVC occurs frequently also in immunocompetent and otherwise healthy women [76,77,78,79,80,81,82]. RVVC requires the long-term use of antifungal drugs to prevent it from relapsing [76,77,83]. One study carried out by Ceccarani et al. in 2019 showed a significant reduction of *Lactobacillus* spp. in infected vaginal ecosystems of 18 females suffering from VVC compared to 21 healthy individuals [84]. These findings suggest a protective role of *Lactobacillus* spp., such as *L. crispatus*, to prevent overgrowth of *Candida* spp. However, some studies also suggest an abundance of *Lactobacillus* spp. with a greater incidence of *Candida* colonization [87,88,95]. Eastment et al. conducted a study in 2021 to analyze the microbiota composition of VVC patients in Kenya and the United States. Microbiota composition was acquired using amplicon sequencing. The study demonstrated that a higher relative abundance of *Megasphaera* species and *Mageeibacillus indolicus* was associated with a lower risk of yeast detection in the vaginal area. Moreover, higher abundances of *Bifidobacterium bifidum*, several *Streptococcus* spp., and *Aerococcus christensenii* were correlated with yeast detection, suggesting a possible interaction of those species, thereby increasing the colonization of yeast species in the vaginal area [85]. Other studies suggest the co-occurrence of *Candida* with other bacterial vaginosis (BV)-associated bacteria [95,96,97,98,99].

An important limitation of these studies is that *Lactobacillus* spp. were not identified at the species level [84,87,88,95]. Different *Lactobacillus* spp. can either be beneficial for the host and protect from an infection with *Candida* spp., or increase the risk of such a colonization. Therefore, identification to the species level is crucial when regarding the impact of *Lactobacillus* spp. on candidiasis. For example, studies hypothesized that the presence of *L. iners* in the female reproductive tract creates a suitable environment for the co-occurrence of *C. albicans*, especially when compared to *L. crispatus*. Tortelli et al. proved this hypothesis by examining the vaginal microbiome of women. They observed that women whose reproductive tract were colonized by *L. iners* had an increased incidence of the co-occurrence of *Candida* spp. in comparison to women with diverse vaginal microbiomes and those colonized by *L. crispatus*. It was speculated that *L. iners* produces more lactic acid, which increases the pH of the vagina and significantly supports the colonization and co-occurrence of *Candida* spp. [93]. However, a main limitation of this study was that the entire vaginal microbiome may not have been captured because of the use of amplicon sequencing only to detect *Lactobacillus* species. [93]. The microorganisms, which might play a role in *C. albicans* colonization and infection, are summarized in Table 2.

There is a high possibility that other anaerobic bacteria present in the female reproductive tract may play crucial roles in the etiopathogenesis of VVC and RVVC. The authors also did not consider the various stages of the menstrual cycles of the women sampled. It would be interesting to understand the role of the menstrual cycle in the initiation and progression of VVC and if the vaginal microbiota composition changes during the different stages of the menstrual cycle. A study using modern shotgun genome sequencing would shed more light on these interesting research questions. With regard to *Candida* balanitis, a lot is still unknown about the influence of the host-microbiome on disease occurrence. Studies carried out by Meng et al. showed that sexual behavior (the use of condoms), circumcision, lifestyle, and personal hygiene play an important role in disease incidence. They observed that men who use condoms are more likely to have *Staphylococcus* spp. as the prevalent bacteria implicated in disease severity, while men without sexual activities are more likely to harbour *Prevotella* spp. [100]. However, considerably less studies have been carried out on *Candida* balanitis than vaginitis.

### 4.3. Psoriasis

Psoriasis (PS) is a chronic inflammatory skin disease thought to have a genetic and immunological link [101]. About 2 percent of the world’s population is affected by PS. Its prevalence varies according to geographical locations [101,102]. Studies have also shown that Caucasians and Scandinavians are more likely to be affected by PS than Africans and Asians [102,103,104,105]. The etiology of PS remains vividly debated; however, some studies have shown a correlation between the skin microbiota dysbiosis and the occurrence of PS [101,106]. Several studies have established a relationship between the colonization of the psoriatic skin lesions and *S. aureus*. The authors observed that approximately 60 percent of people affected by psoriasis have their skin and nares colonized by *S. aureus*, as compared to 3 to 30 percent in healthy people, thereby pointing out the possibility that *S. aureus* may exacerbate PS flares [106,107,108]. Viruses, such as the human papillomaviruses (HPV) have also been implicated to play a role in the occurrence of PS [106]. Fry and Baker proposed that HPV infection of the keratinocyte is supported by epidermal proliferation, and that epidermal hyperproliferation, which is a characteristic of PS, results in an opportunistic infection [106]. Arguably, HPV might be one assumed autoantigen recognized by the CD4^+^ and CD8^+^ T cells in psoriatic lesions [106,109]. *Malassezia* spp. have also been hypothesized as contributors in the development of this disease. *Malassezia* is a common inhabitant of the scalp [106]. It was observed that orally administered antifungal drugs resulted in a significant decrease in the abundance of the yeast on the scalp and an improvement of the disease condition [110,111]. To further examine this finding, Lober et al. carried out patch testing of sonicated, heat-killed *Malassezia* spp. cells on patients with inactive PS. It was observed that skin lesions clinically and histologically similar to PS were formed [110]. To further demonstrate the relationship between the skin microbiota and PS flares, Alekseyenko et al. performed a study between 2008 and 2011, which included 75 patients with psoriasis and 124 healthy controls. Skin swabs were obtained from all participants. Diseased individuals were swabbed at the affected and unaffected body sites. DNA was extracted and the bacterial composition analyzed by amplicon sequencing. The authors found a significant increase in the abundance of *Corynebacterium*, *Propionibacterium*, *Staphylococcus*, and *Streptococcus* in psoriatic plaques [55]. In contrast to the previous results, Gao et al. revealed that members of the phylum *Firmicutes* were overrepresented in PS lesions, while the members of the phylum *Actinobacteria* and *Cutibacterium* spp. were the least abundant [57]. Likewise, Quan et al. analyzed the microbiota of healthy individuals and affected and unaffected skin of PS patients. They observed an abundance of *Corynebacterium* sp. and a decrease in the abundance of *Cutibacterium* in PS lesions [112]. Drago et al. identified a dysbiosis of the microbiome of one Italian PS patient in 2016. They observed that a significant increase in the abundance of Proteobacteria and a decrease in the abundance of *Streptococcaceae, Rhodobacteraceae, Campylobacteraceae, Moraxellaceae,* and *Firmicutes* was associated with PS plagues [113]. Contrary to the result of the experiment published by Lober et al., Paulino revealed that the occurrence and distribution of *Malassezia* spp. were not host-specific and its population on the skin remained stable over time. Additionally, there was no significant difference in the abundance of *Malassezia* in both diseased and healthy skin areas of PS patients. However, only two individuals, one male and one female located in North America were included in the study [114]. In order to estimate a significance of these findings, more participants should be included in the study. Different climate zones should also be considered, as the environment (e.g., temperature range, humidity, and air pollution) plays a major role in the skin microbial composition [115]. A summary of all microorganisms mentioned above, which might play a role in disease severity and/or progression, is shown in Table 3.

Another study from Italy employed shotgun metagenomic sequencing to analyze the skin microbiome of PS patients. Skin lesions were compared to non-lesional skin areas. The findings suggested an increase in *Staphylococcus* spp. in lesioned skin (Table 3). On average, the observed alpha diversity, a measure to analyze the biodiversity of microbiomes, was significantly lower in diseased compared to unaffected areas [115]. Similar results were shown concerning the gut microbiome of PS patients. Indeed, an interesting study by Todberg and colleagues analyzed very recently whether PS also has an impact on the gut microbiome. They performed shotgun metagenomic sequencing on fecal samples of 53 untreated patients with plaque PS and 52 healthy controls. Overall, participants suffering from plaque PS showed a significantly reduced alpha diversity of their gut microbiomes compared to healthy participants. They further confirmed the higher abundance of *Blautia* spp. in diseased participants, whereas *Faecalibacterium* spp. displayed a lower abundance [116]. The limitations of the aforementioned studies include small sample sets, sometimes a lack of precise details on the actual study design, and the more commonly used amplicon sequencing instead of untargeted sequencing methods.

### 4.4. Correlation of the Skin Microbiome and Other Dermatological Conditions

Apart from the three selected skin conditions, i.e., (i) hidradenitis suppurativa (acne inversa), (ii) (muco-)cutaneous candidiasis, and (iii) psoriasis, there are many other skin conditions, for which the role of the microbiome has been described regarding disease initiation and progression. Atopic dermatitis (AD), for example, has been extensively studied. AD is a chronic inflammatory disease, affecting 1–3% of adults and 5–20% of children worldwide [117,118,119]. Especially an increase of *S. aureus* has been correlated with lesional skin areas of AD patients and a more severe form of the disease [120]. In contrast, *S. epidermidis* has been associated with milder cases of AD [8,121,122]. Overall, a lower alpha diversity has also been shown for AD patients in contrast to healthy skin [118,123,124]. Specific bacterial genera that decreased in lesional skin of AD patients are *Streptococcus*, *Cutibacterium*, and *Corynebacterium* [8,125]. A shift to a more diverse microbiome has previously been achieved after UV-B exposure [126]. However, probiotic baths and lotions, as well as a specific inhibition of *S. aureus* might also become adjunctive therapy options.

Skin cancer, such as malignant melanoma and basal cell carcinoma, has also been shown to be affected by the cutaneous microbiome [127,128,129]. One striking study carried out by Mekadim et al. in 2022 detected an increase in *Fusobacterium*, *Trueperella*, *Staphylococcus*, *Streptococcus*, and *Bacteroides* in melanoma tissue, compared to healthy skin. The researchers were also able to show a two-fold decrease in alpha diversity of melanoma tissue in contrast to healthy skin [126].

Next to skin cancer, certain autoimmune diseases also play a crucial role in dermatology, e.g., systemic sclerosis and bullous pemphigoid. The skin microbiome has been described for both dermatological conditions [130,131]. Johnson et al. have analyzed the skin microbiome of systemic sclerosis patients in 2019. They described an increase in gram-negative taxa, such as *Burkholderia*, *Citrobacter*, and *Vibrio*, whereas lipophilic taxa seem to be increased [131]. More research is needed to evaluate existing microbiome data and gain new insights into the complex interplay of microbes and the host.

## 5. Conclusions and Perspectives

There is an evident association and possibly a causal relationship between the skin microbiota and the initiation, maintenance, and progression of several dermatological pathologies, which warrants further investigation. Although a lot is still unknown about the molecular mechanisms giving rise to these interactions, it is becoming increasingly evident that an in-depth understanding of these associations will be key to the development of new therapeutic and, potentially, preventive strategies. Thus far, it is acknowledged that several pathologies are associated with an increase or decrease in certain microorganisms. However, as many of the affected bacteria are common commensals of the human skin, an accurate understanding will require further typing beyond the sole species level to understand how specific bacteria interact within the human body. Recent advances in metagenomics and computational biology will enable us to decipher these intricate relationships. Ideally, future research should focus on skin microbiota-derived metabolites and how they can be exploited as therapeutic tools. In contrast to standard microbiological diagnostic methods, which only allow the detection of cultivatable strains or search for specific strains performing polymerase chain reaction, next generation sequencing techniques, in particular shotgun metagenomic sequencing of whole-genome DNA extracted from clinical samples enable the analysis of all microorganisms present. This can help with drawing correlations of certain, thus far not disease-associated species with (non-)infectious skin conditions. In the case of psoriasis, acne inversa and candidiasis, therapy could benefit from readjusting a balanced skin microbiome by using probiotic treatment which could help curing a potential microbiome dysbiosis. In that regard, it will also be interesting to elucidate the effects of several non-invasive dermatological procedures (e.g., UV light therapy) on the skin microbiota. Beneficial metabolites, produced by the microorganisms on our skin are another option for an improved therapy strategy. Many therapeutics currently used to treat skin diseases have side effects, e.g., (i) antibiotics, (ii) antifungals, and (iii) immunosuppressants. Microbiota-derived metabolites have a decreased risk of causing side effects since they are synthesized by microorganisms living on our skin and we are exposed to them and the compounds they produce on a daily basis. Most microorganisms constantly compete with each other for nutrients and space, thus being an excellent target for searching novel antimicrobial compounds. They also harbor the potential to be species-specific, meaning these metabolites inhibit the growth of or act bactericidal towards only a few, rather than all other microorganisms inhabiting the skin. Besides methodological biases (e.g., due to amplicon sequencing), many of the studies published thus far are limited by the relatively small sample sizes. Hence, it appears promising to comparatively analyze the skin microbiome of large cohorts across diverse geographical locations, to fully understand the effect of climate, lifestyle, age, sex, and other variable factors on the microbial diversity of the skin.

## Figures and Tables

**Table 1 life-12-01420-t001:** **Summary of microorganisms that change abundance in hidradenitis suppurativa lesions (acne inversa).** List of microorganisms, which may be either increased or decreased in abundance during hidradenitis suppurativa. An increase is marked with an arrow pointing up, a decrease with an arrow pointing down, and contradictory results are marked with one arrow pointing left and one pointing right.

Microorganism	Abundance
*Staphylococcus aureus*	
*Streptococcus pyogenes*	
*Pseudomonas aeruginosa*	
*Peptostreptococcus* spp.	
*Prevotella* spp.	
Microaerophilic *Streptococcus* spp.	
*Fusobacterium* spp.	
*Cutibacterium acnes*	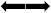
*Staphylococcus lugdunensis*	
*Porphyromonas* spp.	
*Aerococcus* spp.	
*Mobiluncus* spp.	

**Table 2 life-12-01420-t002:** **Summary of microorganisms that change abundance during (muco-)cutaneous Candidiasis.** List of microorganisms, which may be either increased or decreased in abundance during hidradenitis suppurativa. An increase is marked with an arrow pointing up, a decrease with an arrow pointing down, and contradictory results are marked with one arrow pointing left and one pointing right.

Microorganism	Abundance
*Staphylococcus aureus*	
*Streptococcus* spp.	
*Aerococcus christensenii*	
*Lactobacillus iners*	
*Prevotella* spp.	
*Megasphaera* spp.	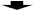
*Mageeibacillus indolicus*	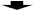
*Lactobacillus crispatus*	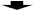
*Bifidobacterium bifidum*	

**Table 3 life-12-01420-t003:** **Summary of microorganisms that change abundance in psoriasis lesions.** List of microorganisms, which may be either increased or decreased in abundance during hidradenitis suppurativa. An increase is marked with an arrow pointing up, a decrease with an arrow pointing down, and contradictory results are marked with one arrow pointing left and one pointing right.

Microorganism	Abundance
*Staphylococcus aureus*	
*Malassezia* spp.	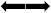
*Corynebacterium* spp.	
*Firmicutes*	
*Proteobacteria*	
*Cutibacterium spp.*	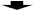
*Actinobacteria*	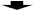

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
