# Peer review of "Characteristics of the Skin Microbiome in Selected Dermatological Conditions: A Narrative Review"

_life, 2022, doi:10.3390/life12091420_

Round 1
Reviewer 1 Report
General comments
The article entitled “Characteristics of the skin microbiome in selected dermatological conditions: a narrative review” review some common diseases (acne inversa, candidiasis, pso- 22 riasis) associated with dysbiosis of microbial communities and highlights points about microbiome sequencing approaches.
Abstract
Please include in the text how recent research approaches used in skin microbiome studies may lead to improved diagnostics and therapeutics in the foreseeable future. This point was not discussed but only cited in the conclusion section.
1. Introduction
Lines 75-76: “shotgun metagenomic sequencing (1-3,8). Here, all the genetic materials”. Please consider changing “here” to “Through shotgun metagenomic sequencing”, or “Through shotgun technique”, or similar.
3. The role of microbiota in healthy skin section
The section is more about skin composition and microbe-microbe interaction than the role of the microbiota in healthy skin. Aspects like the most abundant organisms, the stability of skin microbial communities despite constant environmental changes, or the distinct adaptations to survive on the skin could be addressed.
Lines 110-111: Please explain how the skin plays a vital role in stimulating and training the human immune system. Lunjani and colleagues (ref. 38) cite two papers to comment that “changes in environment, lifestyle and dietary factors may play a role in the miseducation or deficient training of the immune system.”
4.1. Hidradenitis suppurativa (acne inversa) section
Table 1, Table 2, and Table 3: The arrows appear to be dividing the two columns of the table, not indicating results. Show numerical data followed by references from where these abundance data were obtained is more relevant than the arrows.
4.2. (Muco-)Cutaneous Candidiasis
As the manuscript proposes to review skin microbiome this section should address cutaneous candidiasis exploring the microbiota adhered to the skin instead of vulvovaginal candidiasis. Please consider rewriting this section or changing the manuscript title.
Lines 242-243: please discuss why the non-identification of Lactobacillus spp. to the species level is an important limitation of these studies. And specify which studies are referred to.
Author Response
Please see our detailed point-by-point response in the attached document.

Reviewer 2 Report
The manuscript „CHARACTERISTICS OF THE SKIN MICROBIOME IN SELECTED DERMATOLOGICAL CONDITIONS: A NARRATIVE REVIEW “ presents various aspects of skin microbiota of three skin conditions. This manuscript is valuable as it presents useful information and a review on this specific subject, based on the wide literature data and analysis of clinical study results.
However, I consider that the manuscript should be improved and I have some suggestions:
The authors talk about 3 different diseases/conditions, so please add an explanation why choose them and possible reasons, if any. If you talk about common skin conditions, hidradenitis is not so common. So I suggest you mention/write psoriasis in the first position.
-TITLE: I consider that you may mention the names of your three diagnoses in the title, to be more specific for readers.
-KEYWORDS: I consider that you must mention the names of your three diagnoses, to be more specific and for potential further recognitions/citations.
IN THE TEXT:
- The authors write T-cells and T cells . However, I suggest using T cells.
- The authors write “Arguably, HPV is the assumed autoantigen recognized 289 by the CD+T cells in psoriatic lesions (106,108).” Which CD+ number?
- The authors write “However, only two individuals, one male 316 and one female located in North America were included in the study (113).” What does this mean for the subject – general knowledge on skin microbiome?
Finally, at the end of the manuscript you can mention other skin diseases descibed by articles which more widely present available data on skin microbiome – for example you can see the article from LIFE journal: Ferček et al., 2021. Features of the Skin Microbiota in Common Inflammatory Skin Diseases. Life (Basel). 2021 Sep 14;11(9):962. doi: 10.3390/life11090962.
However, since I am a dermatologist, I can comment on the text as a clinician, and the suggestions by microbiologists or molecular biologists would be welcome.
Reviewer 3 Report
interesting manuscript but please see all my comments in the text directly and correct / complete to be acceptable for publication

Round 2
Reviewer 2 Report
The text of the manuscript is improved.
Reviewer 3 Report
no further corrections are necessary. my comments have been well taken into account